# Abnormalities of the Halogen Bonds in the Complexes between Y_2_CTe (Y = H, F, CH_3_) and XF (X = F, Cl, Br, I)

**DOI:** 10.3390/molecules27238523

**Published:** 2022-12-03

**Authors:** Ya-Qian Wang, Rui-Jing Wang, Qing-Zhong Li, Zhi-Wu Yu

**Affiliations:** 1MOE Key Laboratory of Bioorganic Phosphorous Chemistry and Chemical Biology, Department of Chemistry, Tsinghua University, Beijing 100084, China; 2The Laboratory of Theoretical and Computational Chemistry, School of Chemistry and Chemical Engineering, Yantai University, Yantai 264005, China

**Keywords:** halogen bond, hydrogen bond, abnormality, competition, AIM, NBO

## Abstract

In this work, the hydrogen bonds and halogen bonds in the complexes between Y_2_CTe (Y = H, F, CH_3_) and XF (X = F, Cl, Br, I) have been studied by quantum chemical calculations. We found three interesting abnormalities regarding the interactions. Firstly, the strength of halogen bonds increases in the order of IF < BrF < ClF < F_2_. Secondly, the halogen bonds formed by F_2_ are very strong, with an interaction energy in the range between −199.8 and −233.1 kJ/mol. Thirdly, all the halogen bonds are stronger than the hydrogen bonds in the systems we examined. All these results are against the general understanding of halogen bonds. These apparent abnormal properties are reconciled with the high polarizability of the Te atom and the strong inducing effect of F on the Te atom of Y_2_CTe. These findings provide a new perspective on halogen bonds. Additionally, we also proposed bonding distance-based methods to compare the strength of halogen/hydrogen bonds formed between different donor atoms and the same acceptor atom.

## 1. Introduction

A halogen bond is formed between an electrophilic region of a halogen atom X (X = F, Cl, Br, and I) in a molecule R-X (R is an electron-withdrawing atom/group) and a nucleophilic region of a molecule Y-R’ [1], denoted as R-X∙∙∙Y-R’. The electrophilic region, or the electron-deficient region of the X atom, is located along the R-X σ-bond, denoted as σ-hole, which is surrounded by a belt of negative electrostatic potential [2]. Nowadays, halogen bonds have received extensive attention due to their important roles in many fields such as supramolecular chemistry, organocatalysis, synthetic coordination chemistry, polymer chemistry, and drug discovery [3,4,5,6,7,8,9,10,11,12,13]. For example, halogen bonding has been a popular and much exploited supramolecular synthon in the crystal field [5,9]. The application of halogen atoms as pharmaceutically active ligand substituents has been widespread in recent medicinal chemistry [10,11].

The properties of halogen bonds are related to their strength, which is not only dependent on the halogen donor atom and the acceptor atom, but is also affected by substituents. Normally, the halogen bond becomes stronger with the halogen donor varying from F to I [1,12,13,14,15,16]. An electron-donating group in the halogen bond acceptor strengthens the halogen bond, while an electron-withdrawing group in the acceptor has a weakening effect [16]. The type of the halogen bond acceptor varies from anions and neutral molecules with lone pairs to π-electron molecules, radicals, metal hydrides, and carbenes [17,18]. Specially, the molecules containing N and O atoms are often taken as the halogen bond acceptor.

It is interesting to study the differences between hydrogen bonds and halogen bonds, since both types of interactions have comparable strength and may coexist in the same systems [19,20,21,22,23,24,25,26,27]. Usually, hydrogen bonds are stronger than halogen bonds, except for when an iodine atom acts as the halogen donor [24,28]. Thus, some studies have tried to make halogen bonds stronger than hydrogen bonds [28,29,30]. When the halogen bond acceptor H_2_CO binds with the hydrogen/halogen donor HOBr, the interaction energy of the hydrogen bond is larger by 7 kJ/mol than that of the halogen bond [19]. Inversely, the interaction energy of the halogen bond is larger by 1 kJ/mol than that of the hydrogen bond if H_2_CO is changed to H_2_CS [28]. This difference is enlarged to 2 kJ/mol when one H atom of H_2_CS is replaced by a Li atom [28]. These results indicate that the differences between hydrogen bonds and halogen bonds can be regulated by changing the halogen bond acceptor atoms and/or adding substituents. Nevertheless, these comparisons are not very convincing, because, for HOBr to participate in hydrogen bonding and halogen bonding interactions, the remaining moieties are -OBr and -OH, respectively, meaning that they are not identical. To overcome this difficulty, Li and coauthors designed a molecule called 6-OX-fulvene (X = H, Cl, Br, I), where the moiety of fulvene increases the acidity of the X atom. Then, they examined the interactions between this molecule and ZH_3_/H_2_Y (Z = N, P, As, and Sb; Y = O, S, Se, and Te) [31]. It was found that the hydrogen bond is weakened with the Lewis base atom growing in size; however, the effect of the same on halogen bonds is very limited [31]. If SbH_3_ and H_2_Te are selected as the acceptors, the halogen bonds are much stronger than the hydrogen bonds, and the largest difference in their interaction energies is 40 kJ/mol in the SbH_3_∙∙∙6-OCl-fulvene complex [31].

H_2_CTe is a homologue of H_2_CO and H_2_CS; thus, it can also work as an acceptor to form hydrogen bonds or halogen bonds. Considering that Te is a semimetal located on the dividing line between metals and non-metals, we expect that the halogen bond formed by it may have different patterns. In this study, we investigated the complexes between Y_2_CTe (Y = H, F, and CH_3_) and XF (X = H, F, Cl, Br, and I), wherein XF is a hydrogen/halogen bond donor and Y_2_CTe is an acceptor. With the strong electronegativity of F, the designated molecules XF are expected to be prominent halogen bond donors. The following questions are addressed by the method of quantum chemical calculations: (1) Whether the halogen bond is stronger than the hydrogen bond. (2) Whether the strength of halogen bond follows the order of F_2_ < ClF < BrF < IF. (3) What is the nature of the hydrogen bond and halogen bond in these complexes?

## 2. Results

### 2.1. Molecular Electrostatic Potential (MEP) Analyses

It is well known that the MEP diagram of a molecule is helpful to effectively predict noncovalent interactions involving that molecule [32]. Figure 1 shows the MEP maps of two families of molecules: Y_2_CTe (Y = F, H, and CH_3_) and XF (X = H, F, Cl, Br, and I). The MEP distributions in both families are anisotropic. For Y_2_CTe, we focus on the negative areas of the MEPs (blue colored areas). As expected, there are mainly two negative areas in each molecule, which correspond to the lone pairs of the Te atom. Compared with H_2_CTe (−78.8 kJ/mol), the minimal MEP value of the Te atom decreases in F_2_CTe (−52.5 kJ/mol) but increases in (CH_3_)_2_CTe (−99.8 kJ/mol), which can be attributed to the electron-withdrawing nature of F atoms and the electron-donating ability of the methyl groups, respectively.

For XF, we focus on the positive areas of the MEPs (red areas). In the case of HF, the atom H exhibits positive electrostatic potential, while F is negative. For dihalogen molecules XF, there is a positive MEP region (σ-hole) at the X atom along the X-F bond. The magnitude of the σ-hole on the halogen atom increases with an increasing atomic mass of X. It is also found that the maximal MEP on the H atom is larger than that on the halogen atoms, including iodine.

### 2.2. Geometries

For the hydrogen bonding or halogen bonding interactions (Y_2_CTe∙∙∙XF), the general geometry of the complexes and the involved parameters are shown in Figure 2. We focus on the Te∙∙∙X distance (R_1_), the change in the X-F bond length (∆R_2_), and the Te∙∙∙X-F angle (α). The data of the optimized structures are listed in Table 1.

As can be seen in the table, all the values of R_1_ are much shorter than the sum of the van der Waals radii of the respective atoms (3.3 Å for Te and H, 3.6 Å for Te and F, 4.0 Å for Te and Cl, 4.2 Å for Te and Br, and 4.4 Å for Te and I) [33,34]. This justifies the formation of hydrogen/halogen bonds. Further, the interactions between the electron-donor and acceptor molecules seem to be quite strong because stronger interaction is known to result in shorter bond length (R_1_). To compare the relative strength between the halogen bonds between different interaction partners and the hydrogen bond, we define a quantity ΔR_1_% in the following equation: (1)$${\Delta R}_{1}\%=\frac{R_{c}-R_{1}}{R_{c}}\times 100\%$$
where R_c_ is the sum of the van der Waals radii of the two atoms representing the critical distance to judge the presence of a hydrogen/halogen bond. After normalization with R_c_, the shortening of the Te∙∙∙X distance could be used to evaluate the strength of hydrogen/halogen bonds. Thus, for each of the three molecules (H_2_CTe, F_2_CTe, and (CH_3_)_2_CTe), the ΔR_1_% are all in the sequential order F_2_ > ClF > BrF > IF > HF when they form interaction pairs. This implies that all the halogen bonds are stronger than the hydrogen bonds. Most interestingly, the ΔR_1_% values suggest that the halogen bond strength decreases with an increasing size of the halogen atom in the donor molecule XF. This is different from the general understanding of halogen bonds.

The change in the X-F bond length R_2_ of a donor could also reflect the interaction strength of the hydrogen/halogen bond. Here, we calculated the change of R_2_ relative to the R_2_ in the monomer, denoted as ΔR_2_%, using the following formula:(2)$${\Delta R}_{2}\%=\frac{{\Delta R}_{2}}{R_{2}}\times 100\%$$
The ΔR_2_% represents the elongation percentage of the X-F bond, and the larger ΔR_2_% implies more significant weakening of the bond and, thus, stronger interaction. As indicated by ΔR_2_%, the value of X-F bond length is larger in the halogen-bonded complex than that in the hydrogen-bonded analogue. This relative elongation in the halogen-bonded complex decreases in the order of F_2_ > ClF > BrF > IF. These data are supportive of the conclusions from ΔR_1_%.

The Te∙∙∙X-F angle (α) is in the range of 168–180°, confirming a good direction of the hydrogen/halogen bonds. The angles are less than 180° in the majority of the complexs due to the attraction between the Y atom/group in Y_2_CTe and XF.

### 2.3. Energies

Here, we consider the interaction energy to be the most credible criteria to judge the strength of interactions. Therefore, we calculated the interaction energies (E_int_) of the various complexes for comparing the hydrogen and the halogen bonds. We used the counterpoise correction method to eliminate the basis set superposition error (BSSE), and the corrected energy is denoted as E_int,BSSE_. In addition, the more accurate energy E_int,CBS,BSSE_ with complete basis set (CBS) was also calculated. The results with and without BSSE correction, as well as with CBS, are all listed in Table 2. The main concern of our study is that the changing trends of the interaction energy with the variation of X in XF are the same based on all the three methods. It is worth clarifying that the following discussions about energies in the full text are all according to their absolute values. As shown in Table 2, the interaction energies of hydrogen bonds in all of the three series of complexes are smaller than those of the halogen bonds, indicating that the hydrogen bonds are weaker than all of the halogen bonds. For the strength order of the halogen bonds, both E_int,BSSE_ and E_int,CBS,BSSE_ increased in the order of IF < BrF < ClF < F_2_ for the series of H_2_CTe∙∙∙XF and (CH_3_)_2_CTe∙∙∙XF complexes. This result is abnormal compared to the common perception that the halogen bond becomes stronger with the halogen donor varying from F to I. For F_2_CTe∙∙∙XF complexes, the E_int,BSSE_/E_int,CBS,BSSE_ of ClF, BrF, and IF was close. For all of the three series of the complexes, F_2_ molecules formed the strongest halogen bonds. The absolute values of the interaction energy were very large, up to 228.8 kJ/mol for E_int,BSSE_ and 233.1 kJ/mol for E_int,CBS,BSSE_. To compare the interaction energies of the halogen bonds formed by different acceptors (Y_2_CTe), when Y is F, an electron-withdrawing atom, the Y_2_CTe∙∙∙XF interaction, was weakened and compared to that of H_2_CTe. On the contrary, when Y was the electron-donating methyl group, the interaction was strengthened.

To understand the attribution of the interaction energy, we partitioned it into five terms: electrostatic energy (E^es^), exchange energy (E^ex^), repulsion energy (E^rep^), polarization energy (E^pol^), and dispersion energy (E^disp^), and the data are listed in Table 3. Obviously, E^ex^ is the largest attractive term in each complex; thus, it plays the most important role in the stabilization of hydrogen/halogen bonds [35,36]. This term increases in the order of HF < IF < BrF < ClF < F_2_, which is consistent with the results of orbital interaction discussed in the following section. The large E^ex^ of each complex suggests a strong orbital interaction between the two respective monomers. For the repulsive term, E^rep^ was very large and even exceeds 1000 kJ/mol in each F_2_-related complex. This can be attributed to the much shorter Te∙∙∙X distances. It is seen that E^rep^ was almost twice as much as E^ex^ and both terms have a good linear relationship (Appendix A), confirming their dependency each other. 

Now, we examine the three attractive terms (E^es^, E^pol^, and E^disp^) in Table 3 in some detail (intuitively in Appendix A). For the hydrogen bond complex Y_2_CTe∙∙∙HF, E^es^ was the largest attractive interaction among the three terms, followed by E^pol^. For the interaction energies of halogen bonds formed by ClF, BrF, or IF with all the three acceptors Y_2_CTe, the contributions of electrostatic and polarization interactions are comparable. However, for the interaction energies of the halogen bonds formed by F_2_ with the Y_2_CTe, the polarization interaction is the dominating contribution. This may be attributed to the special property of F, namely the largest electronegativity in the periodic table, thus possessing a very strong inducing ability.

### 2.4. Atoms in Molecules (AIM) Analyses

The hydrogen/halogen bonds can be characterized by the Te∙∙∙X bond critical points (BCPs, Appendix A). The most important properties of each bond critical point are summarized in Table 4, where *ρ* refers to the electron density, ∇^2^*ρ* its Laplacian, and *H* the energy density [37,38,39]. Generally, the larger electron density *ρ* reflects the stronger interaction. For all the investigated systems, *ρ* increases in the sequential order of HF < IF < BrF < ClF < F_2_, in agreement with the order of interaction energies (Table 2). For the Laplacian, it was seen that ∇^2^*ρ* > 0 for all the complexes, demonstrating that the interactions studied were closed shell interaction. The energy density *H* is a more sensitive parameter than ∇^2^*ρ*. The negative values of *H* further demonstrate that the interactions are partially covalent in nature. In the complexes involving (CH_3_)_2_CTe, there are also BCPs between the methyl H and the halogen X in HX or XFs (Appendix A), indicating the presence of C-H∙∙∙F/X hydrogen bonds. The interaction energies of the C-H∙∙∙F/X hydrogen bonds were estimated by E = 0.5V(r) [40,41], where V(r) is the potential energy density at a BCP in each case. The corresponding data are −5.6, −17.8, −10.3, −9.9, and −8.1 kJ/mol for (CH_3_)_2_CTe∙∙∙HF, (CH_3_)_2_CTe∙∙∙F_2_, (CH_3_)_2_CTe∙∙∙ClF, (CH_3_)_2_CTe∙∙∙BrF, and (CH_3_)_2_CTe∙∙∙IF, respectively. Clearly, these hydrogen bonds contributed to the stability of the complexes; however, their shares in the total interaction energies (Table 2) are small. Subtracting them from the total interaction energies, the residual results still have the same change trend with the total interaction energy. Thus, the presence of C-H∙∙∙F/X hydrogen bonds does not affect the abnormality of halogen bonds.

### 2.5. Natural Bond Orbital (NBO) Analyses

The charge transfers (CTs) from Y_2_CTe to XF are listed in Table 5, which are calculated as the sum of the charge on all atoms in Y_2_CTe in the complexes. The charge transfer reflects the electrons bias from electron donor (Y_2_CTe) to electron acceptor (XF), providing information about the interaction strength in one aspect. As can be seen, the CTs are all larger than 0.2 e in the halogen bonds and even exceed 0.8 e in the F_2_ complexes. On the contrary, the hydrogen-bonded complexes have much smaller CTs than the halogen-bonded analogues. Additionally, for all three of the series of complexes, the CT value decreases in the order F_2_ > ClF > BrF > IF > HF, which is the same as the interaction strength order. Besides, when the Y is the electron-withdrawing atom F in Y_2_CTe, the CT becomes smaller than that in H_2_CTe. When the Y is the electron-donating methyl group, the CT increases.

There is an orbital interaction between the lone pair orbital on the Te atom of Y_2_CTe and the anti-bonding orbital of the X-F bond (Lp_Te_→σ^*^x-_F_), and this orbital interaction can be measured with the second-order perturbation energy (E^2^), which is also listed in Table 5. This orbital interaction is not detected in the F_2_-containing complexes since the F-F bond is taken as two subunits in the NBO analysis. The E^2^ has a consistent change order with the charge transfer. This orbital interaction is strong; therefore, it makes an important contribution to the formation of hydrogen/halogen bond. We also calculated the dipole moments of the complexes (Table 5). It was found that the order of the dipole moment is consistent with the interaction energy. Further, the relationship between the CTs and the population of the σ*x-_F_ orbital was analyzed, and positive correlation was found (Appendix A). This suggests that the main destination/receiver of the CT is the σ*x-_F_ orbital in each complex.

## 3. Discussion

Hydrogen bonds and halogen bonds are two important noncovalent interactions, and they often coexist; thus, it is interesting to compare their strength. Generally speaking, hydrogen bonds are considered to be stronger than halogen bonds [28]. Interestingly, in the present study, using Y_2_CTe (Y = H, F, and CH_3_) as electron donors, we found that their halogen bonding interactions with dihalogen molecules XF (X = F, Cl, Br, and I) are stronger than their hydrogen bonding interactions with HF. This apparent abnormality was also seen in a previous study on the interactions between PH_3_/AsH_3_/H_2_Te and 6-OX-fulvene (X = H, Cl, Br, I) [31]. The second abnormality found in this work was that, when the X changes from F to I, the halogen bond becomes weaker, in contrast to the normal understanding that stronger halogen bonds accompany heavier halogen donors [16]. The abnormalities can be explained by the high polarizability of the Te atom, which is easily polarized when the electronegative XF approaches it. The greater the electronegativity of the approaching atom, the greater the polarization of the Te atom. Therefore, when the X atom of XF varies from I to F, the dipole moment of the complex increases, as seen in Table 5, and the polarization energy (the major contribution to the interaction energy) also increases. Based on the data in Table 3 and Table 5, a near positive correlation between the polarization energy and the dipole moment of the complex is found (Appendix A). 

Another surprising result was that F_2_ participates in the strongest halogen bond with the interaction energy up to −233.1 kJ/mol in the (CH_3_)_2_CTe∙∙∙F_2_ complex. Such large interaction energy is abnormal because it shows the least MEP at the end of the X atom among the four XF molecules. The apparent contradiction can be reconciled as follows. The F atom of F_2_, due to it having the highest electronegativity among the halogens, would cause the largest polarization on the Te atom and, thus, the largest dipole moment of the Y_2_CTe∙∙∙XF complex. This is evidenced by the largest polarization energies being seen in the three Y_2_CTe∙∙∙F_2_ complexes. The polarization mechanism is also consistent with the charge transfer data, which are the biggest in the Y_2_CTe∙∙∙F_2_ complexes, even as big as 0.943e in the complex of (CH_3_)_2_CTe∙∙∙F_2_. Such big charge transfers (>0.8e) mean that the molecule F_2_ holds nearly an extra electron in each of the three complexes, similar to the process of becoming an anion.

In order to test if the above abnormal results are found only for Y_2_CTe, the Te atom of H_2_CTe was replaced by O, S, and Se. The corresponding interaction energies are listed in Table 6. For the lighter chalcogen atoms O and S, the halogen bond strength increases in the order of F_2_ < ClF < BrF < IF, which is the “normal” understanding of halogen bonds. For the heavier Se as the electron donor, the halogen bonds formed by ClF, BrF, and IF turned out to be comparable, while that formed by F_2_ increased tremendously. This complicated situation is explained as follows. On one hand, a Se atom with a larger radius is more easily polarized than O and S. On the other hand, it is not as easily polarized as Te. Thus, only the most electronegative F_2_ is able to assert marked influence on the electron distribution of H_2_CSe, making it the strongest interaction in the H_2_CSe∙∙∙XF complexes. 

As discussed above, the size/polarizability of the chalcogen atoms in H_2_CZ (Z = O, S, Se, and Te) plays a very important role in the strength of halogen bonds. The data in Table 6 demonstrate that, for a given XF (X = F, Cl, Br, and I), the strength of its halogen bond with H_2_CZ increases monotonously when Z varies from O to Te. On the contrary, the strength of the hydrogen bond between HF and H_2_CZ decreases monotonously. As a result, for H_2_CO, only IF forms a stronger halogen bond than the hydrogen bond. For H_2_CS, each dihalogen molecule, excluding F_2_, participates in a stronger halogen bond than the hydrogen bond formed by HF. For H_2_CSe and H_2_CTe, all the dihalogen molecules form stronger halogen bonds than hydrogen bonds formed by HF.

The above discussions about the interaction strength of halogen/hydrogen bonds are based on their interaction energies, often regarded as golden criteria. Practically, other parameters such as electrostatic potential (MEP), halogen/hydrogen bond length (R_1_), and donor bond length (R_2_) are often used to compare the interaction strength. MEP sometimes provides correct indications. For example, the MEP order is Cl < Br < I at the end of X atom in XF, and the halogen bond strength order is ClF < BrF < IF in the F_2_CTe∙∙∙XF complexes. However, the F has the least MEP at the end of the X atom among the four XF molecules, but it forms the strongest interaction with F_2_CTe. Clearly, MEP cannot always provide correct results because it only contains the information of isolated molecules. For R_1_ and R_2_, due to the different radii of the halogen atoms, we defined two quantities, ΔR_1_% and ΔR_2_%, to compare the halogen/hydrogen bond strength. The correlations between the two quantities and the respective interaction energies are plotted in Figure 3 (the trend comparisons are shown in Appendix A). Undoubtedly, they are all positively correlated, suggesting that both ΔR_1_% and ΔR_2_% can be used to compare the halogen/hydrogen bond strength qualitatively. Quantitatively, ΔR_2_% is a better choice.

## 4. Computational Methods

All calculations were carried out with the Gaussian 09 program [42]. The geometries of all the monomers and complexes were optimized at the MP2 level with the aug-cc-pVTZ basis set for all atoms except I and Te, where the aug-cc-pVTZ-PP basis set was adopted to account for relativistic corrections [43]. For all atoms in all the complexes, collectively, aug-cc-pVTZ(PP) represents the basis set used in this work. The extrema of molecular electrostatic potentials (MEPs) were calculated on the 0.001 a.u. isodensity surface at the MP2/aug-cc-pVTZ(PP) level using the WFA-SAS program [32]. The interaction energy (E_int_) of a complex was computed as the difference between the energy of the complex and the sum of the energies of the monomers with their geometries frozen in the optimized complex. For this supermolecular method of calculating the MP2 interaction energy, the dispersion correction was taken into account, since MP2 contains certain correlation terms such as the uncoupled Hartree-Fock (UCHF) dispersion energy, the corresponding Hartree-Fock exchange-dispersion energy, and a deformation-correlation term [44]. Interaction energies were corrected for basis set superposition error (BSSE) using the counterpoise procedure (CP) proposed by Boys and Bernardi [45]. The two-point extrapolated energies with a complete basis set (CBS) proposed by Halkier et al. [46,47] were obtained with two basis sets of aug-cc-pVDZ(PP) and aug-cc-pVTZ(PP). The localized molecular orbital-energy decomposition analysis was used to decompose the interaction energy into five terms of electrostatic, exchange, repulsion, polarization, and dispersion at the MP2/aug-cc-pVTZ(PP) level with the GAMESS program [48]. The dispersion energy was obtained as a difference between the MP2 and CCSD(T) energy in the GAMESS program. The AIM2000 package [49] was used to assess the topological parameters at each bond critical point (BCP), including electron density, as well as its Laplacian, and energy density. Using the natural bond orbital (NBO) method [50] within the Gaussian 09 program, the charge transfer and second-order perturbation energy were obtained.

## 5. Conclusions

Quantum chemical calculations have been performed to study the interactions between Y_2_CTe (Y = H, F, and CH_3_) and XF (X = H, F, Cl, Br, and I). The results show that the electron-withdrawing groups F in F_2_CTe weaken the interactions, while the electron-donating methyl groups in (CH_3_)_2_CTe strengthen them. More importantly, we found three abnormalities regarding halogen bonds in this work. The first one is that the strength of halogen bond increases in the sequential order IF < BrF < ClF < F_2_ in H_2_CTe∙∙∙XF and (CH_3_)_2_CTe∙∙∙XF complexes. This is contrary to the normal understanding that the stronger halogen bonds accompany heavier halogen donors. The second one is that the halogen bonds formed by F_2_ are very strong, even up to −233.1 kJ/mol with (CH_3_)_2_CTe. Contrary to this, the halogen bonds formed by F-R are normally considered to be very weak or even negligible. The last one is that all halogen bonds are stronger than the hydrogen bonds in the complexes, which is abnormal compared with the majority of previous studies. These abnormalities are discussed in the context of the high polarizability of the Te atom in the halogen acceptors. Because the Te atom is easily polarized when the electronegative XF approaches it, the greater the electronegativity of the X atom, the greater the polarization of the Te atom. Particularly, the F atom has the largest electronegativity in the periodic table and possesses a very strong inducing ability. Consequently, F-F forms tremendously strong interactions with Y_2_CTe.

## Figures and Tables

**Figure 1 molecules-27-08523-f001:**
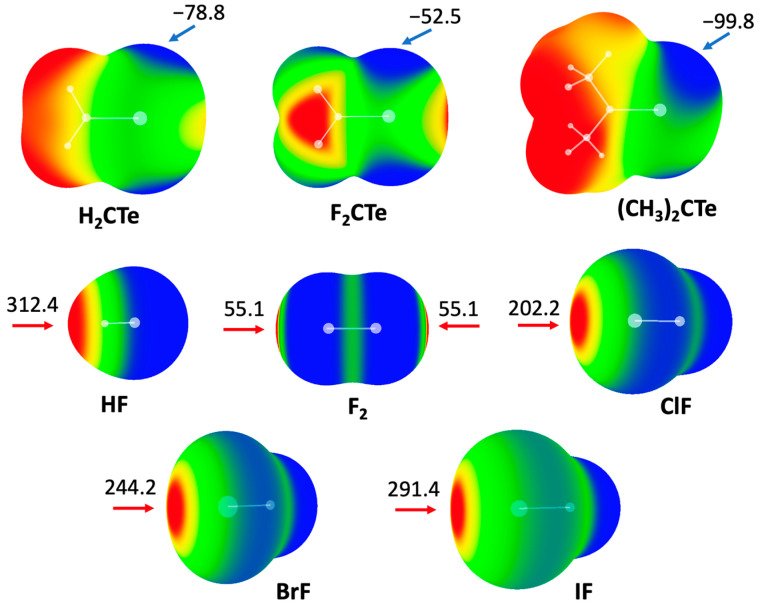
MEP diagrams of molecules studied in this work. Color ranges, in kJ/mol: red, greater than 52.5; yellow, between 52.5 and 0; green, between 0 and −52.5; blue, less than −52.5. Arrows refer to values of maxima and minima.

**Figure 2 molecules-27-08523-f002:**
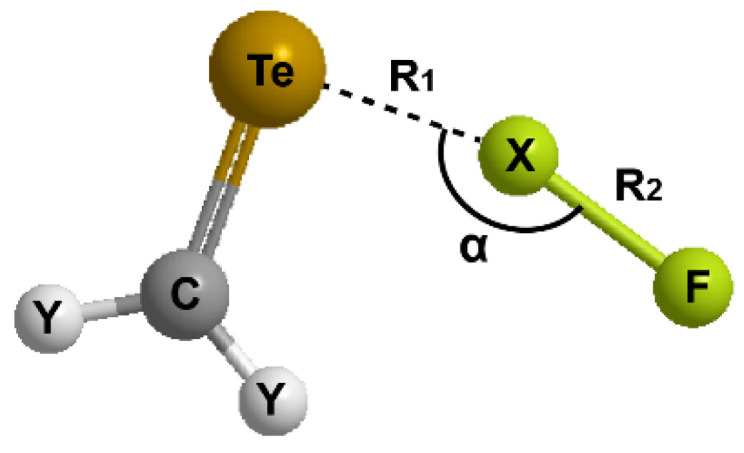
Illustration of the general structure of Y_2_CTe∙∙∙XF complex.

**Figure 3 molecules-27-08523-f003:**
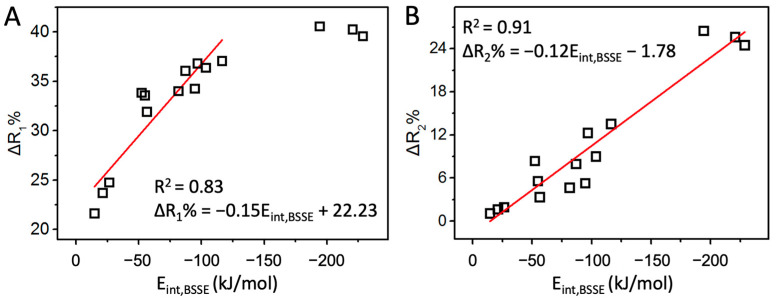
The correlation of E_int,BSSE_ with ΔR_1_% (**A**) and ΔR_2_% (**B**).

**Table 1 molecules-27-08523-t001:** Binding distance (R_1_, Å), ΔR_1_%; elongation of the X-F bond length (ΔR_2_, Å), ΔR_2_%; bond angle (α, deg) of the complexes.

	R_1_	ΔR_1_%	ΔR_2_	ΔR_2_%	α
H_2_CTe∙∙∙HF	2.518	23.70%	0.015	1.63%	168.6
H_2_CTe∙∙∙F_2_	2.151	40.25%	0.359	25.62%	168.8
H_2_CTe∙∙∙ClF	2.528	36.80%	0.201	12.26%	176.8
H_2_CTe∙∙∙BrF	2.686	36.05%	0.140	7.96%	177.4
H_2_CTe∙∙∙IF	2.904	34.00%	0.089	4.64%	177.9
F_2_CTe∙∙∙HF	2.586	21.64%	0.010	1.08%	179.9
F_2_CTe∙∙∙F_2_	2.140	40.56%	0.371	26.48%	170.7
F_2_CTe∙∙∙ClF	2.647	33.83%	0.137	8.36%	179.7
F_2_CTe∙∙∙BrF	2.790	33.57%	0.098	5.57%	179.8
F_2_CTe∙∙∙IF	2.996	31.91%	0.064	3.33%	179.2
(CH_3_)_2_CTe∙∙∙HF	2.483	24.76%	0.018	1.95%	170.7
(CH_3_)_2_CTe∙∙∙F_2_	2.176	39.56%	0.343	24.48%	168.0
(CH_3_)_2_CTe∙∙∙ClF	2.518	37.05%	0.222	13.54%	178.1
(CH_3_)_2_CTe∙∙∙BrF	2.673	36.36%	0.158	8.99%	179.1
(CH_3_)_2_CTe∙∙∙IF	2.893	34.25%	0.101	5.26%	180.0

**Table 2 molecules-27-08523-t002:** Interaction energies (E_int_) corrected with and without BSSE in the complexes at the MP2/aug-cc-pVTZ(PP) level, all in kJ/mol.

		HF	F_2_	ClF	BrF	IF
H_2_CTe	E_int_	−27.8	−232.6	−107.0	−100.0	−94.8
	E_int,BSSE_	−21.1	−220.7	−96.7	−87.0	−81.5
	E_int,CBS,BSSE_	−22.3	−225.5	−106.5	−95.6	−89.9
F_2_CTe	E_int_	−21.1	−206.9	−61.2	−66.4	−68.8
	E_int,BSSE_	−14.5	−194.4	−52.7	−54.8	−56.4
	E_int,CBS,BSSE_	−15.6	−199.8	−60.5	−61.7	−63.6
(CH_3_)_2_CTe	E_int_	−33.7	−242.2	−128.1	−120.0	−111.8
	E_int,BSSE_	−26.3	−228.8	−116.4	−103.6	−94.7
	E_int,CBS,BSSE_	−27.7	−233.1	−126.5	−112.5	−103.3

**Table 3 molecules-27-08523-t003:** Electrostatic (E^es^), exchange energy (E^ex^), repulsion energy (E^rep^), polarization (E^pol^), and dispersion (E^disp^) energies in the complexes, all in kJ/mol.

	E^es^	E^ex^	E^rep^	E^pol^	E^disp^
H_2_CTe∙∙∙HF	−33.0	−48.9	90.4	−21.3	−8.3
H_2_CTe∙∙∙F_2_	−246.0	−519.1	1097.3	−442.5	−110.3
H_2_CTe∙∙∙ClF	−255.5	−474.9	985.5	−278.4	−73.4
H_2_CTe∙∙∙BrF	−251.3	−417.7	865.0	−223.0	−60.1
H_2_CTe∙∙∙IF	−183.2	−346.0	678.3	−182.8	−47.8
F_2_CTe∙∙∙HF	−22.8	−38.4	70.8	−16.7	−7.4
F_2_CTe∙∙∙F_2_	−242.6	−524.2	1109.5	−423.0	−114.1
F_2_CTe∙∙∙ClF	−170.5	−344.4	695.7	−169.2	−63.9
F_2_CTe∙∙∙BrF	−171.7	−312.3	630.0	−148.2	−52.6
F_2_CTe∙∙∙IF	−129.4	−265.6	510.3	−129.3	−42.3
(CH_3_)_2_CTe∙∙∙HF	−46.2	−64.5	118.7	−26.2	−8.1
(CH_3_)_2_CTe∙∙∙F_2_	−253.4	−533.8	1116.9	−457.1	−101.5
(CH_3_)_2_CTe∙∙∙ClF	−287.7	−522.2	1080.9	−315.4	−72.1
(CH_3_)_2_CTe∙∙∙BrF	−287.6	−468.2	966.8	−253.4	−61.2
(CH_3_)_2_CTe∙∙∙IF	−211.6	−391.0	763.3	−205.4	−50.0

**Table 4 molecules-27-08523-t004:** Electron density (*ρ*), Laplacian (∇^2^*ρ*), and total energy density (*H*) at the intermolecular BCP in the complexes (all values in a.u.).

	*ρ*	∇^2^*ρ*	*H*
H_2_CTe∙∙∙HF	0.022	0.032	−0.003
H_2_CTe∙∙∙F_2_	0.095	0.101	−0.036
H_2_CTe∙∙∙ClF	0.079	0.024	−0.027
H_2_CTe∙∙∙BrF	0.066	0.033	−0.020
H_2_CTe∙∙∙IF	0.052	0.041	−0.013
F_2_CTe∙∙∙HF	0.019	0.032	−0.001
F_2_CTe∙∙∙F_2_	0.096	0.112	−0.036
F_2_CTe∙∙∙ClF	0.060	0.058	−0.015
F_2_CTe∙∙∙BrF	0.052	0.053	−0.012
F_2_CTe∙∙∙IF	0.042	0.049	−0.008
(CH_3_)_2_CTe∙∙∙HF	0.025	0.031	−0.004
(CH_3_)_2_CTe∙∙∙F_2_	0.091	0.088	−0.033
(CH_3_)_2_CTe∙∙∙ClF	0.082	0.013	−0.029
(CH_3_)_2_CTe∙∙∙BrF	0.069	0.024	−0.022
(CH_3_)_2_CTe∙∙∙IF	0.054	0.036	−0.015

**Table 5 molecules-27-08523-t005:** Charge transfer (CT, e), second-order perturbation energy (E^2^, kJ/mol), and dipole moment (μ, D) in the complexes.

	CT	E^2^	μ
H_2_CTe∙∙∙HF	0.043	70.0	3.17
H_2_CTe∙∙∙F_2_	0.893	-	10.52
H_2_CTe∙∙∙ClF	0.511	781.6	8.23
H_2_CTe∙∙∙BrF	0.414	761.4	7.87
H_2_CTe∙∙∙IF	0.322	636.6	7.81
F_2_CTe∙∙∙HF	0.031	51.0	2.88
F_2_CTe∙∙∙F_2_	0.843	-	9.53
F_2_CTe∙∙∙ClF	0.334	462.3	5.62
F_2_CTe∙∙∙BrF	0.293	417.7	5.93
F_2_CTe∙∙∙IF	0.242	334.3	6.34
(CH_3_)_2_CTe∙∙∙HF	0.056	93.3	4.76
(CH_3_)_2_CTe∙∙∙F_2_	0.943	-	11.73
(CH_3_)_2_CTe∙∙∙ClF	0.579	711.7	10.37
(CH_3_)_2_CTe∙∙∙BrF	0.462	696.9	9.96
(CH_3_)_2_CTe∙∙∙IF	0.350	613.1	9.84

**Table 6 molecules-27-08523-t006:** Interaction energies (E_int,BSSE_, kJ/mol) of complexes between H_2_CZ (Z = O, S, Se, and Te) and XF (X = H, F, Cl, Br, and I) at the MP2/aug-cc-pVTZ(PP) level.

	HF	F_2_	ClF	BrF	IF
H_2_CO	−34.4	−5.9	−25.2	−34.4	−42.3
H_2_CS	−26.6	−7.3	−52.2	−60.6	−64.0
H_2_CSe	−24.6	−171.7	−64.7	−68.2	−69.6
H_2_CTe	−21.1	−220.7	−96.7	−87.0	−81.5

## Data Availability

Not applicable.

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
