# Peer review of "Abnormalities of the Halogen Bonds in the Complexes between Y_2_CTe (Y = H, F, CH_3_) and XF (X = F, Cl, Br, I)"

_molecules, 2022, doi:10.3390/molecules27238523_

Round 1

Reviewer 1 Report

This paper studied the interplay between Y2CTe (Y = H, F, CH3) and XF (X = F, Cl, Br, I) in function of intermolecular distance, interaction energy, the LMOEDA scheme, NBO theory and AIM topological parameters. The work could be of great interest in the field of halogen bonds because the conclusions are against the general understanding of these interactions. However, the entire article needs a thorough review before being considered for publication in Molecules.

Some suggestions:

1) Although the calculation methodology is suitable for geometric optimizations. The authors should consider refining the energetic calculations, for example using the extrapolation scheme proposed by Halkier et al. following the recommendations of Jurecka et al. (J. Chem. Phys. 111 (1999) 9157–9167, Chem. Phys. Lett. 302 (1999) 437–446, Phys. Chem. Chem. Phys. 8 (2006) 1985–1993), or some other extrapolation scheme.

2) When the authors carry out the energetic decomposition (LMOEDA), they do not take into account the exchange energy (Eex: is the most stabilizing of all). There are recent works that consider that exchange/correlation-exchange plays an important role in the stabilization of hydrogen bonds (J Comput Chem. 2021;1–13) and halogen bonds (Chemical Physics Letters 710 (2018) 113–117, ChemistrySelect 2021, 6, 680–684 680).

3) QTAIM analysis offers a wide variety of parameters that are very useful for characterizing molecular interactions. However, this work does not take advantage of them.

4) Lately, for large sets of basis functions, interaction energies are not corrected for basis set superposition error (BSSE).

5) Line 219: The authors state “Therefore, when the X atom of XF varies from I to F, the dipole moment of the complex increases as seen in Table 5, and the polarization energy (the major contribution to the interaction energy) also increases. However, the reader does not have any figure to see how the magnitude of the vector varies. Perhaps, the authors can make use of the atomic dipole moment (QTAIM), or of the Laplacian topology of the electron density, to justify this statement.

 6) Lines 298-308: The calculation levels indicated are not correct or are not indicated.

7) Lines 83-89: Electrostatic potential values must be in kJ/mol

8) Line 92: The magnitude of σ-hole on the halogen atom not only depends on the atomic mass of this.

9) Table 2 and Figure 3 must be unified.

10) Line 231: the authors say “Such big charge transfers mean that the F2 in Y2CTe∙∙∙F2 complexes are nearly ionized”. They should discuss this result, with the experimental value of the ionization energy of the F2 molecule. Also, an atomic charge analysis could be useful in this case.

Author Response

The Comments of Reviewer 1 and Our Reponses:

This paper studied the interplay between Y2CTe (Y = H, F, CH3) and XF (X = F, Cl, Br, I) in function of intermolecular distance, interaction energy, the LMOEDA scheme, NBO theory and AIM topological parameters. The work could be of great interest in the field of halogen bonds because the conclusions are against the general understanding of these interactions. However, the entire article needs a thorough review before being considered for publication in Molecules.

Comment 1: Although the calculation methodology is suitable for geometric optimizations. The authors should consider refining the energetic calculations, for example using the extrapolation scheme proposed by Halkier et al. following the recommendations of Jurecka et al. (J. Chem. Phys. 111 (1999) 9157–9167, Chem. Phys. Lett. 302 (1999) 437–446, Phys. Chem. Chem. Phys. 8 (2006) 1985–1993), or some other extrapolation scheme.

Response: Thanks for the comments and suggestion. We have calculated the interaction energies using the two-point extrapolation scheme with two basis sets of aug-cc-pVDZ(PP) and aug-cc-pVTZ(PP), and the data are listed in the Table 2. Respective references as suggested have also been added to the manuscript.

The following sentence has been added:

“The two-point extrapolated energies with complete basis set (CBS) proposed by Halkier et al [43-45] were obtained with two basis sets of aug-cc-pVDZ(PP) and aug-cc-pVTZ(PP).” (In section “Computational Methods”)

Comment 2: When the authors carry out the energetic decomposition (LMOEDA), they do not take into account the exchange energy (Eex: is the most stabilizing of all). There are recent works that consider that exchange/correlation-exchange plays an important role in the stabilization of hydrogen bonds (J Comput Chem. 2021;1–13) and halogen bonds (Chemical Physics Letters 710 (2018) 113–117, ChemistrySelect 2021, 6, 680–684 680).

Response: Thanks for the comments. The energy decomposition in our work considered exchange energy. They were listed in Table 3 but without description. Now we have added the discussion about the exchange energy in section 2.3 as follows: “Obviously, Eex is the largest attractive term in each complex, thus it plays the most important role in the stabilization of hydrogen/halogen bonds [35,36]. This term increases in the order of HF < IF < BrF < ClF < F2, which is consistent with the results of orbital interaction discussed in the following section. The large Eex of each complex suggests the strong orbital interaction between the two respective monomers.”

In addition, the two literature works have been added as references 35 and 36.

Comment 3: QTAIM analysis offers a wide variety of parameters that are very useful for characterizing molecular interactions. However, this work does not take advantage of them.

Response: In section 2.4, we have utilized the data of AIM analysis to characterize the hydrogen and halogen bonds including their strength and nature.

Comment 4: Lately, for large sets of basis functions, interaction energies are not corrected for basis set superposition error (BSSE).

Response: We performed BSSE correction in our work. Now we have added the interaction energies without BSSE and with the correction using CBS, which are also listed in Table 2. The following sentences have been added in section 2.3: “We used the counterpoise correction method to eliminate the basis set superposition error (BSSE), the corrected energy is denoted as Eint,BSSE. In addition, the more accurate energy Eint,CBS,BSSE with complete basis set (CBS) was also calculated. The results with and without BSSE correction, as well as with CBS are all listed in Table 2. For the main concern of our study, the changing trends of the interaction energy with the variation of X in X-F are the same based on all the three methods.”

Comment 5: Line 219: The authors state “Therefore, when the X atom of XF varies from I to F, the dipole moment of the complex increases as seen in Table 5, and the polarization energy (the major contribution to the interaction energy) also increases. However, the reader does not have any figure to see how the magnitude of the vector varies. Perhaps, the authors can make use of the atomic dipole moment (QTAIM), or of the Laplacian topology of the electron density, to justify this statement.

Response: We added Figure S4 in the supporting information to describe the correlation between the dipole moment and the polarization energy. The following sentences have been added: “Based on the data in Tables 3 and 5, a near positive correlation between the polarization energy and the dipole moment of the complex is found (Figure S4).” (In section 3)

Comment 6: Lines 298-308: The calculation levels indicated are not correct or are not indicated.

Response: Thanks! The basis set for iodine was missed. Actually, aug-cc-pVTZ-PP was used not only for tellurium but also for iodine. For all the other atoms, the basis set was aug-cc-pVTZ. The respective sentences are revised as: “The geometries of all the monomers and complexes were optimized at the MP2 level with the aug-cc-pVTZ basis set for all atoms except I and Te, where the aug-cc-pVTZ-PP basis set was adopted to account for relativistic corrections [41]. For all atoms in all the complexes collectively, aug-cc-pVTZ(PP) represents the basis set used in this work.”

Comment 7: Lines 83-89: Electrostatic potential values must be in kJ/mol.

Response: This has been corrected.

Comment 8: Line 92: The magnitude of σ-hole on the halogen atom not only depends on the atomic mass of this.

Response: We agree to this point that the magnitude of σ-hole on a halogen atom of a molecule not only depends on the atomic mass of this, it is also influenced by the substituent on the halogen atom. But in our study the substituent (F) is fixed, leaving the atomic mass as the only influencing factor. The sentence in section 2.1 has been modified as: “The magnitude of the σ-hole on the halogen atom increases with increasing atomic mass of X.” Another sentence is added to the last paragraph of the Introduction section: “With the strong electronegativity of F, the designated molecules X-F are expected to be prominent halogen bond donors.”

Comment 9: Table 2 and Figure 3 must be unified.

Response: Figure 3 was deleted since the corresponding data have been given in Table 2.

Comment 10: Line 231: the authors say “Such big charge transfers mean that the F2 in Y2CTe∙∙∙F2 complexes are nearly ionized”. They should discuss this result, with the experimental value of the ionization energy of the F2 molecule. Also, an atomic charge analysis could be useful in this case.

Response: Thanks a lot for the comment. The sentence is not properly expressed and may cause misunderstanding. It has been revised as: “Such big charge transfers (> 0.8e) mean that the molecule F2 holds nearly an extra electron in each of the three complexes, like becoming an anion.”

Reviewer 2 Report

The manuscript by Ya-Qian Wang and co-workers reports on theoretical research the hydrogen bonds and halogen bonds in the complexes between Y2CTe (Y = H, F, CH3) and XF (X = F, Cl, Br, I). The work used an arsenal of modern theoretical methods: MEP, QTAIM, NBO, and EDA. The authors identified a number of features that contradict the generally accepted understanding of the halogen bond. The paper reveals regularities between distance and interaction energy. It should be noted that the work has similarities in methodology with the work of S. Scheiner and co-workers. (10.1002/cphc.201900340) In general, the work is well written and structured. However, more careful analysis and extension is needed to make the results more reliable and useful in studies of non-covalent interactions. After that, the manuscript can be recommended for publication in the journal Molecules.

1.       The presence of a halogen bond between Y2CTe and F2 is doubtful and debatable. (See work 10.1007/s00894-012-1591-0) It is necessary to carry out in more detail in the text the evidence of attribution to the halogen bond and explain your point of view.

2.       In Fig. 1, the mapping of H2CTe, F2, and (СH3)2CTe molecules should be corrected.

3.       Since there is an anomaly in the regularity (IF < BrF < ClF < F2 in H2CTe∙∙∙XF), the interaction energy calculated with and without BSSE error should be analyzed.

4.       In my opinion, a number of adducts (СH3)2CTe∙∙∙XF were not chosen well, since in addition to the halogen bond X∙∙∙Te there is a hydrogen bond X∙∙∙H (as evidenced by the presence of a path and a BCP point between the halogen and hydrogen atom on the molecular graph of Fig. S2). However, the text does not mention this, and it is not clear whether the correction for hydrogen bonding X∙∙∙H was taken into account.

5.       Fig 4 does not show the error for the linear relationship.

6.       Analyzing the results of the EDA analysis, the authors focus on the energies Ees, Epol, and Edisp and point out the dominant role of the polarization forces of the Y2CTe∙∙∙F2 interaction. However, the repulsion energy, which is important in the formation of interactions between molecules, is not taken into account. The role of the Erep in the formation of interactions should be explained in the text.

7.       To get a better idea of charge transfer, the results of the NBO analysis should be supplemented with the population of the σ*x-F orbitals. Does the same correlation hold with NBO charges?

8. It is not clear from the methodology how the dispersion correction was taken into account. This should be explained in detail

Author Response

The Comments of Reviewer 2 and Our Responses:

The manuscript by Ya-Qian Wang and co-workers reports on theoretical research the hydrogen bonds and halogen bonds in the complexes between Y2CTe (Y = H, F, CH3) and XF (X = F, Cl, Br, I). The work used an arsenal of modern theoretical methods: MEP, QTAIM, NBO, and EDA. The authors identified a number of features that contradict the generally accepted understanding of the halogen bond. The paper reveals regularities between distance and interaction energy. It should be noted that the work has similarities in methodology with the work of S. Scheiner and co-workers. (10.1002/cphc.201900340) In general, the work is well written and structured. However, more careful analysis and extension is needed to make the results more reliable and useful in studies of non-covalent interactions. After that, the manuscript can be recommended for publication in the journal Molecules.

Comment 1: The presence of a halogen bond between Y2CTe and F2 is doubtful and debatable. (See work 10.1007/s00894-012-1591-0) It is necessary to carry out in more detail in the text the evidence of attribution to the halogen bond and explain your point of view.

Response: Thanks for the comment. The halogen bond studied in the literature mentioned above is CF3-F∙∙∙NH3, where the N atom in NH3 is the halogen bond acceptor. However, in our work, the halogen bond acceptor is the larger-sized atom Te in Y2CTe, which has higher polarizability than N atom. When the electronegative donor atom F approaches to the Te atom, the Te atom will be polarized and forms a strong interaction with F-F. When the halogen bond acceptors are atoms without high polarizability, such as O and S, as shown in Table 6 in the manuscript, they form weak interactions with F2. Practically, the most popular criterion to judge the presence of hydrogen/halogen bonds in a complex is to compare the donor-acceptor distance with the sum of the vdW radii of the concerned atoms. In our work, the binding distances R1 (Te∙∙∙F-F) are in the range of 2.1 ~2.2 Å, much shorter than the sum of the van der Waals radii of the atoms Te and F (3.6 Å). Additionally, the optimized structures Y2CTe∙∙∙F2 show that the interaction directions are along the σ-bond of F2, and there exist bond critical points between Te and F (Figure S2). Based on these reasons, we conclude that halogen bonds form between Y2CTe and F2.

Comment 2: In Fig. 1, the mapping of H2CTe, F2, and (СH3)2CTe molecules should be corrected.

Response: Thanks. The missing covalent bonds between atoms of H2CTe, F2, and (СH3)2CTe molecules in Figure 1 are now added. The unit of the potential has been changed from a.u. to kJ/mol.

Comment 3: Since there is an anomaly in the regularity (IF < BrF < ClF < F2 in H2CTe∙∙∙XF), the interaction energy calculated with and without BSSE error should be analyzed.

Response: The interaction energies without BSSE are added to Table 2. The following sentences have been added in section 2.3: “We used the counterpoise correction method to eliminate the basis set superposition error (BSSE), the corrected energy is denoted as Eint,BSSE. In addition, the more accurate energy Eint,CBS,BSSE with complete basis set (CBS) was also calculated. The results with and without BSSE correction, as well as with CBS are all listed in Table 2. For the main concern of our study, the changing trends of the interaction energy with the variation of X in X-F are the same based on all the three methods.”

Comment 4: In my opinion, a number of adducts (СH3)2CTe∙∙∙XF were not chosen well, since in addition to the halogen bond X∙∙∙Te there is a hydrogen bond X∙∙∙H (as evidenced by the presence of a path and a BCP point between the halogen and hydrogen atom on the molecular graph of Fig. S2). However, the text does not mention this, and it is not clear whether the correction for hydrogen bonding X∙∙∙H was taken into account.

Response: Thanks! We have added the following discussion in the section 2.4: “In the complexes involving (CH3)2CTe, there are also BCPs between the methyl H and the halogen X in HX or XFs (Figure S2), indicating the presence of C-H∙∙∙F/X hydrogen bonds. The interaction energies of the C-H∙∙∙F/X hydrogen bonds were estimated by E = 0.5V(r), where V(r) is the potential energy density at a BCP in each case. The corresponding data are -5.61, -17.76, -10.31, -9.86, and -8.05 kJ/mol for (CH3)2CTe∙∙∙HF, (CH3)2CTe∙∙∙F2, (CH3)2CTe∙∙∙ClF, (CH3)2CTe∙∙∙BrF, and (CH3)2CTe∙∙∙IF, respectively. Clearly, these hydrogen bonds have contributions to the stability of the complexes. But their shares in the total interaction energies (Table 2) are small. Subtracting them from the total interaction energies, the residual results have still the same change trend with the total interaction energy. Thus the presence of C-H∙∙∙F/X hydrogen bonds does not affect the abnormality of halogen bonds.”

Comment 5: Fig 4 does not show the error for the linear relationship.

Response: We have added the R2 values of the linear relationship in Figure 4.

Comment 6: Analyzing the results of the EDA analysis, the authors focus on the energies Ees, Epol, and Edisp and point out the dominant role of the polarization forces of the Y2CTe∙∙∙F2 interaction. However, the repulsion energy, which is important in the formation of interactions between molecules, is not taken into account. The role of the Erep in the formation of interactions should be explained in the text.

Response: Thanks for the suggestion. We have added the discussion about the repulsion energy in section 2.3 as follows: “For the repulsive term, Erep is very large and even exceeds 1000 kJ/mol in each F2-related complex. This can be attributed to the much short Te∙∙∙X distances. It is seen that Erep is almost twice as much as Eex and both terms have a good linear relationship, confirming their dependency each other.”

Comment 7: To get a better idea of charge transfer, the results of the NBO analysis should be supplemented with the population of the σ*x-F orbitals. Does the same correlation hold with NBO charges?

Response: Thanks. We added the correlation between the charge transfer and the population of the σ*x-F orbitals in Figure S3 in the supporting information. Clearly, the population of the σ*x-F orbitals is positively correlated with the charge transfer. The following sentences have been added into section 2.5 of the manuscript: “Further, the relationship between the CTs and the population of the σ*x-F orbital was analyzed and positive correlation was found (Figure S3). This suggests that the main destination/receiver of the CT is the σ*x-F orbital in each complex.”

Comment 8: It is not clear from the methodology how the dispersion correction was taken into account. This should be explained in detail.

Response: The MP2 method has considered the dispersion correction in the Gaussian program. In the Gamess calculations, the dispersion energy is obtained as a difference between the MP2 and CCSD(T) energy. The following sentence has been added in section “Computational Methods”: “The dispersion energy was obtained as a difference between the MP2 and CCSD(T) energy in the GAMESS program.”

Round 2

Reviewer 1 Report

Dear Editor,

The new version of the manuscript has improved remarkably. In my opinion it is possible to publish it in Molecule, after some minor revisions:

1) In some places of the manuscript XF is observed and in others X-F. I think they should unify.

2) Energy values in kJ/mol should have a single decimal place. Unify throughout in the text, figures and tables

3) Lines 151-153: In the sentence “For the strength order of the halogen bonds, both Eint,BSSE and Eint,CBS,BSSE increase in the order of IF < BrF < ClF < F2 for the series of H2CTe···XF and (CH3)2CTe···XF complexes.” Not true in all cases.

4) Lines 173-175: In the sentence “It is seen that Erep is almost twice as much as Eex and both terms have a good linear relationship, confirming their dependency each other.” I think that the authors could facilitate the reading by providing complementary information to the readers, for example, by putting the corresponding figure in the supporting information.

5) Line 199-201: In the sentence “The interaction energies of the C-H···F/X hydrogen bonds were estimated by E = 0.5V(r), where V(r) is the potential energy density at a BCP in each case.” The corresponding reference should be provided.

6) References [43-45] are not correctly indicated.

7) Lines 338-340:  In the sentence “The dispersion energy was obtained as a difference between the MP2 and CCSD(T) energy in the GAMESS program” is not corrected (see for instance J. Chem. Phys. 131, 014102, 2009)

Finally, I consider that this manuscript will be of great impact on this topic. Future revisions are not necessary.

Author Response

The Comments of Reviewer 1 and Our Reponses:

The new version of the manuscript has improved remarkably. In my opinion it is possible to publish it in Molecule, after some minor revisions:

Comment 1: In some places of the manuscript XF is observed and in others X-F. I think they should unify.

Response: Thanks. When we emphasize the molecule, we write as XF. When we emphasize the bond, we write as X-F. Based on the above criteria, we revised all the related expressions.

Comment 2: Energy values in kJ/mol should have a single decimal place. Unify throughout in the text, figures and tables

Response: All the energy values have been unified in this work.

Comment 3: Lines 151-153: In the sentence “For the strength order of the halogen bonds, both Eint,BSSE and Eint,CBS,BSSE increase in the order of IF < BrF < ClF < F2 for the series of H2CTe···XF and (CH3)2CTe···XF complexes.” Not true in all cases.

Response: The order is true for the two series of H2CTe···XF and (CH3) 2CTe···XF complexes as shown in Table 2. For the series of F2CTe···XF, however, the order is not hold exactly. In this case the corrected energies for the three complexes involving ClF, BrF, IF are close each other. This is the reason we only mentioned the two series in our manuscript regarding the sequential order.

Comment 4: Lines 173-175: In the sentence “It is seen that Erep is almost twice as much as Eex and both terms have a good linear relationship, confirming their dependency each other.” I think that the authors could facilitate the reading by providing complementary information to the readers, for example, by putting the corresponding figure in the supporting information.

Response: We have added the relationship between Eex and Erep, which is shown in Figure S1 in the supporting information.

Figure S1. The relationship between repulsion energy Erep and exchange energy Eex in the complexes between Y2CTe (Y = H, F, and CH3) and XF (X = H, F, Cl, Br, and I).

Comment 5: Line 199-201: In the sentence “The interaction energies of the C-H···F/X hydrogen bonds were estimated by E = 0.5V(r), where V(r) is the potential energy density at a BCP in each case.” The corresponding reference should be provided.

Response: Thanks. The corresponding references have been added as references 40 and 41.

Comment 6: References [43-45] are not correctly indicated.

Response: Thanks. The work of reference 45 is an application by using the extrapolation scheme to calculate interaction energy. We have deleted reference 45. Besides, reference 43 and 44 are changed as 45 and 46.

Comment 7: Lines 338-340: In the sentence “The dispersion energy was obtained as a difference between the MP2 and CCSD(T) energy in the GAMESS program” is not corrected (see for instance J. Chem. Phys. 131, 014102, 2009)

Response: This sentence has been removed and another sentence is added in the calculation method: “For this supermolecular method in calculating the MP2 interaction energy, the dispersion correction has been taken into account since MP2 contains certain correlation terms such as the uncoupled Hartree-Fock (UCHF) dispersion energy, the corresponding Hartree-Fock exchange-dispersion energy, and a deformation-correlation term.[44]” [44] Pitonak, M.; Heßelmann, A. Accurate Intermolecular Interaction Energies from a Combination of MP2 and TDDFT Response Theory. J. Chem. Theory Comput. 2010, 6, 168–178.

Reviewer 2 Report

The authors adequately answered the reviewers’ comments and revised the manuscript. This reviewer believes it is now acceptable in Molecules.

Author Response

Thank you very much. We greatly appreciate your valuable comments and suggestions.